# Biochemical Composition and Alkaline Extraction Optimization of Soluble Bioactive Compounds from the Green Algae *Caulerpa cylindraceae*

**DOI:** 10.3390/md23050208

**Published:** 2025-05-14

**Authors:** Amani Tahar, Haïfa Zghida, Débora Tomazi Pereira, Nathalie Korbee, Helen Treichel, Félix L. Figueroa, Lotfi Achour

**Affiliations:** 1Research Laboratory ‘Bioressources: Integrative Biology & Valorization’ (BIOLIVAL) LR14ES06, Higher Institute of Biotechnology of Monastir (ISBM), University of Monastir, Av, Taher Haddad B.P. n°74, Monastir 5000, Tunisia; amanitahar5@gmail.com (A.T.); haifa_zghida@yahoo.com (H.Z.); lotfiachour@yahoo.fr (L.A.); 2Andalusian Institute of Blue Biotechnology and Development (IBYDA), Experimental Center Grice Hutchinson, University of Malaga, Lomas de San Julián, 29004 Malaga, Spain; nkorbee@uma.es (N.K.); felixlfigueroa@uma.es (F.L.F.); 3Laboratory of Microbiology and Bioprocess, Federal University of Fronteira Sul, Erechim 99700-970, Brazil; helen.treichel@uffs.edu.br

**Keywords:** alkaline hydrolysis, bioactive compounds, *Caulerpa cylindracea*, fatty acid, ultrasound-assisted extraction

## Abstract

There is a growing interest in studying the bioactive compounds of invasive green macroalga *Caulerpa cylindracea* due to their potential biotechnological applications. Algal samples were collected from two sites and seasons. Elemental analysis showed the abundance of carbon in the raw material as a source of carbohydrates. The total protein content in different samples ranged from 8.17 to 9.98%. Total lipids in different samples were around 2%. Fatty acid (FA) results revealed the presence of various types, including omega-3 and omega-6 PUFA. Furthermore, an alkaline hydrolysis optimization using response surface methodology was investigated to extract soluble compounds. It showed that the best combination for polyphenols and ABTS was 12.5% sodium carbonate (SC) at 100 °C for 8 h; however, the best combination for proteins and carbohydrates was 7.5% SC at 100 °C for 5 h. A combination of ultrasound pretreatments was carried out to assess the enhancement of the contents. Thus, an increasing amount was recorded for polyphenols and antioxidant capacity. Ultrasound pretreatment results in decreasing extraction time for all compounds. Results showed that the invasive seaweeds, causing environmental impacts in the Mediterranean Sea, represent an interesting source of bioactive compounds.

## 1. Introduction

Marine organisms provide a wealth of structurally diverse and biologically active metabolites, offering a wide range of interesting biological effects and valuable prospects for drug discovery. Recent studies addressing the rising concern of non-indigenous species have shown that understanding marine bioactive compounds can reveal the potential of these species to invade new ecosystems [1,2,3,4,5,6]. According to a recent review, commercializing natural products from marine bioinvaders may effectively mitigate their impact on marine ecosystems [7].

The non-indigenous green algae *Caulerpa cylindracea* (Sonder) [8] poses significant ecological concerns in the Mediterranean Sea. Its adaptability, rapid cycle growth, and propagation by fragmentation allow the occupation of different habitats, causing different impacts on the ecosystem [9]. Nevertheless, extensive research has been conducted on its potential for nutraceutical and pharmaceutical applications [10]. *Caulerpa* species contain several interesting macromolecules for different applications, including sulfated polysaccharides, lipids, proteins, and essential amino acids [11,12]. They have also been shown to contain different secondary metabolites, such as phenols and flavonoids, known for their antioxidant capacity [13].

Polysaccharides are biomacromolecules representing the main structural elements of cell walls in seaweed [14]. Recently, polysaccharides extracted from marine algae have attracted considerable interest for their biocompatibility, diverse bioactive properties, and potential applications in functional foods, cosmetic ingredients, pharmaceuticals, wound dressing, and structural matrices for tissue engineering [15].

Lipids are essential in macroalgae, playing a key role in algal growth and survival. In addition, these biomolecules have potential applications in different industries, such as nutraceutical, pharmaceutical, and food supplements [16]. Generally, total lipids in macroalgae vary between 1% and 10%, depending upon species [17]. Among lipids found in macroalgae, we can cite polar lipids, neutral lipids, and free fatty acids (FA). The FA are the most interesting group of lipids, which can be divided into three groups: saturated FA, monounsaturated FA (MUFA), and polyunsaturated FA (PUFA). Besides, FA is esterified in neutral and polar lipids, whereas polar lipids are the most important for their unique structure and crucial role in cell communication [18].

Proteins are a primary metabolite that is present in significant amounts in macroalgae. Its content varies among species and environmental conditions and may provide all essential amino acids [19,20]. It has been reported that green algae may contain a considerable content of proteins ranging from 9% to 23% of dry weight [21]. These bioactive components present several health benefits, including antioxidant, anti-inflammatory, and antimicrobial activities; antihypertensive; and numerous advantageous functional properties [20].

Polyphenols are highly hydrophilic compounds characterized by hydroxylated aromatic rings [22]. They represent the various secondary metabolites in plants and exhibit multiple biological activities, including antioxidant, anti-properties, anti-inflammatory, antimicrobial, and anticancer potential [23]. Green algae may contain various polyphenols, such as bromophenols, phenolic acids, and flavonoids, that contribute to their nutritional value and potential health benefits [22,24].

Collecting and recycling, along with measures of protection of native species, has been recently recommended by Rotter et al. as an effective solution for safeguarding affected areas from invasive species [25]. Building on this concept, bioactive metabolites of the non-indigenous algae *C. cylindracea* are investigated to use the species as an eradication strategy, supporting biotechnological and recycling initiatives, particularly for cosmeceutical application. Therefore, the objective of this work was to characterize the dried material of the macroalgae collected from different seasons and different sites, determining the ash content, total lipids, and total proteins, and analyzing the fatty acid composition. Furthermore, this study aimed to optimize the extraction of bioactive compounds with antioxidant capacity using response surface methodology (RSM) and ultrasound-assisted extraction (UAE). Based on the ability of alkaline hydrolysis to destabilize the cell wall, enabling the extraction of cytoplasmic compounds and ensuring high-yield results. Another approach for the second part of this work is using ultrasound as a pretreatment combined with alkaline hydrolysis. As a novel and safe technology, UAE presents numerous advantages concerning environmental and economic viewpoints. Besides, this technique was employed to extract bioactive compounds from different matrices, which enhanced the yield results [26]. This initiative was encouraged by the rising demand for natural bioactive ingredients in developing skincare products and pharmaceutical uses, functional components in the food industry, and other biotechnological applications.

Numerous studies have employed a combination of conventional methods, particularly using alkaline hydrolysis with sodium hydroxide with UAE for bioactive compound extraction from algae. These methods showed improvement in yields of different compounds such as polysaccharides, proteins, and phenolic compounds [27,28,29,30,31]. However, in this study alkaline hydrolysis conditions were optimized first using sodium carbonate as a base, then UAE was used as a pretreatment to enhance cell wall disruption and help the release of bioactive compounds with the best extraction of alkaline hydrolysis.

## 2. Results

### 2.1. Caulerpa cylindracea Biochemical Composition

The composition of ash content, total lipids, total proteins, and organic elements (carbon (C), nitrogen (N), hydrogen (H), and sulfur (S)) is represented in the following figure (Figure 1).

The mineral/ash content in *C. cylindracea* showed a non-significant difference between the three studied samples (*p* < 0.05). Besides, summer samples of both sites presented the highest amount, 26.06 ± 0.39% for CSS1 and 26.51 ± 0.40% for CSS2. However, the winter sample exhibited a lower content than the summer samples (17.16 ± 0.18%).

Regarding the total lipids, the analysis showed that all samples of *C. cylindracea* contain a quantity of lipids greater than 3.00% of dry weight (DW) with no significant differences between summer and winter season samples (CSS1 and CWS1) and between the two harvesting sites in summer (CSS1 and CSS2), with the highest amount for CWS1 (2.86 ± 0.013% DW).

The total protein content determined using the conversion factor (N-prot 5.13) demonstrated that all samples exhibited comparable protein levels with no significant differences. However, the sample from the second site showed a higher content (9.98%) than the summer and winter samples from the first site.

Taking into account the analysis of element composition (C, N, H, and S), it demonstrated a corresponding composition between the three different samples except for S for the CSS1, which exhibited a higher amount (2.65%) in comparison with the two other samples, which represented 0.68% and 0.97% for CWS1 and CSS2, respectively. All the samples showed a considerable amount of carbohydrates (C), around 30.00%. However, the H, N, and S percentages were low. Hence, the Carbon to hydrogen (C:N) ratio showed a higher result (22) for CWS1, while for CSS1 and CSS2, the result showed a similar ratio (17.37 and 16.97, respectively).

### 2.2. Algae Fatty Acid Composition

The analysis of FA in the different *C. cylindracea* samples using gas chromatography showed the presence of various groups of FA (Table 1 and Appendix A). The FA profile revealed the presence of saturated (SFA), monounsaturated (MUFA), and polyunsaturated fatty acids (PUFA). Palmitic acid (C16:0) was detected as the most prevalent SFA in all samples. The most abundant MUFA signaled in the different samples is oleic acid (C18:1), followed by palmitic acid (C16:1). Among the PUFA, the presence of linoleic acid (C18:2 ω6), γ-linolenic acid (C18:3ω6), polyunsaturated omega-6 FA, and α-linolenic acid (ALA) (C18:3 ω3) can be noticed in all samples. Additionally, the results showed the presence of eicosatrienoic acid (C20:3), a polyunsaturated omega-3 FA. Except for the MUFA C22:1 that was not detected in both samples CSS2 and CWS1, all the samples showed a similar composition in different FAs. PUFA detected in other samples were 10.73%, 11.28%, and 9.11% for CSS1, CWS1, and CSS2, respectively. PUFA ω3 was similar to PUFA ω6 for CSS1 and CWS1; however, PUFA ω3 was lower than PUFA ω3 in CSS2.

### 2.3. Conventional Method

The conventional method, which used distilled water at room temperature for 24 h, exhibited a low content of bioactive compounds. Polyphenol content and ABTS content (0.56 ± 0.0007 mg GAE per g DW and 59.05 ± 3.83 µmol TEAC per g DW) were significantly different from the best extraction using alkaline hydrolysis (*p* < 0.05). Protein and carbohydrate contents showed low results compared to alkaline hydrolysis (2.19 ± 0.1 mg BSA eq per g DW and 0.76 ± 0.01 mg glucose eq per g DW).

### 2.4. Response Surface Model for Alkaline Hydrolysis

Different response variables used in both models are represented in Table 2 and Table 3. The polyphenols (mg gallic acid eq (GAE) per g DW), proteins (mg BSA (Bovine Serum Albumin) eq per g DW), carbohydrates (mg glucose eq per g DW), and antioxidant activity ABTS (µmol Trolox eq (TEAC) per g DW) were denoted as Y1, Y2, Y3, and Y4, respectively, and each response was performed in triplicate. Some statistic values were studied and represented in Table 4 to fit the models. Analyzing all these data together allows us to validate the studied RSM design; the F-test was used to assess the significance of regression by comparing each F-test value to the F-table value of these models, F_9-5_ = 4.77 (α < 0.05). Applying the second-order polynomial equation shows high significance and proper fit with the experimental results of polyphenol content, ABTS, proteins, and carbohydrate content and exhibits minimal variation around the mean (Table 2 and Table 3) with R^2^ values of 0.90, 0.93, 0.89, and 0.90, respectively.

### 2.5. Effects of Extraction Variables on Polyphenols

After analyzing the results with the RSM, the equation for obtaining the optimum conditions is represented by the linear equation:Polyphenols = 2.09736 + 0.01160 * X_1_ − 1.13594 * X_1_^2^ + 0.79372 * X_2_ − 0.12674 * X_2_^2^ + 0.21604 * X_3_ − 0.37158 * X_3_^2^ + 0.33713 * X_12_ +0.04896 * X_13_ + 0.21767 * X_23_ (R^2^ = 0.90) 

Taking together the R^2^ (0.90), the low *p*-value (<0.001), and the high t-value (7.57) given by the ANOVA test and the positive coefficient of regression (β = 2.09), the significant relationship between the predictive and the response variables. In addition, the equation showed a positive interaction between the factors. The interaction between sodium carbonate (SC) concentration and temperature (X_1,2_) is given by the example of experiences 3 and 14, where the high-temperature changes maintaining the same concentration of SC and extraction time (Table 2). Also, the interaction between temperature and time (X_2,3_) is noteworthy, and this is the showcase of different experiences (3 and 14, 2 and 13) where changing the extraction temperature with the same SC concentration has affected the content of polyphenols (Table 2). However, the interaction between SC concentration and time has a minimal effect on the extraction results, such as extractions 13 and 14 (Table 2).

According to the Pareto chart of polyphenols, temperature is the most effective factor in the response, with an η^2^ = 40.84%. On the other hand, time and SC concentration are not practical factors (Figure 2). However, the F-test value for the polyphenols response is higher than F_9-5_ -tab (Table 4), which allows the validation of the studied model for this response.

Within the surface plot results, the best extraction of the model exists in the optimal area, which is reached with the maximum temperature, which is 100 °C, and 12.5% SC concentration, and an extraction time of less than 8 h, which could be studied to decrease the time of extraction while conserving the same response.

### 2.6. Effects of Extraction Variables on Antioxidant Activity

Following the analysis of ABTS results utilizing RSM, the model showed a highly significant (*p* < 0.001) regression represented by the following equation:ABTS = 330.8855 + 58.5485 * X_1_ − 84.9180 * X_1_^2^ + 99.5563 * X_2 +_ 3.3621 * X_2_^2^ + 22.9206 * X_3_ − 12.0931 * X_3_^2^ + 27.1134 * X_12_ − 12.7727 * X_13_ + 32.5420 * X_23_ (R^2^ = 0.93) 

Overall, the statistical results, as represented by the ANOVA test, the R-squared value (R^2^), the positive coefficient of regression (β = 330.8855), the low *p*-value (<0.001), and the high t-value (12.62), confirm a positive relationship between the predictive and experimental variables. The equation revealed a highly significant impact of temperature and SC concentration, with *p* < 0.05. Also, a positive interaction between the factors has been demonstrated; the interaction between SC concentration and temperature (X_1,2_) is significant, according to experience results (Table 2); increasing the temperature with the same SC concentration has positively affected the antioxidant activity (examples of experiences 3 and 14, Table 2). On the other hand, increasing the SC concentration, with a maximum temperature of 100 °C, has highly improved the results (examples of experiences 12 and 15, Table 2). Additionally, the interaction between SC and time (X_1,3_) is significant, as evidenced by the high temperature (experiments 13 and 14, Table 2), which has enhanced the phenolic content. However, the interaction between SC concentration and time has a minimal effect on the extraction results, as seen in extractions 13 and 14 (Table 2).

Based on the Pareto analysis, temperature and SC concentration positively affect ABTS activity; however, time is not a significant factor in this response (Figure 3). The temperature presents an intense effect with an η^2^ = 36.97%, followed by the SC effect, with η^2^ = 18.80%. The F-test value of 7.86 (Table 4) permits the validation of the ABTS response.

Within the surface plot results, the best extraction of the model exists in the optimal area, which is reached with the maximum temperature, which is 100 °C, and 12.5% SC concentration, and an extraction time of less than 8 h, which could be studied to decrease the time of extraction while conserving the same response.

### 2.7. Effects of Extraction Variables on Proteins

As stated by the data represented in Table 4, it can be confirmed the second model is fitted for protein response. The F-test value has been improved by modifying the studied SC concentration levels and extraction time. Analyzing the results with the RSM, the equation for obtaining the optimum conditions for proteins is represented by the linear equation:Proteins = 9.51624 + 1.49972 * X_1_ − 7.46312 * X_1_^2^ + 5.548476 * X_2_ + 3.46642 * X_2_^2^ + 1.25728 * X_3_ − 0.97356 * X_3_^2^ + 1.05398 * X_12_ +0.52451 * X_13_ + 1.01704 * X_23_ (R^2^ = 0.89)

In keeping with the R^2^ and the positive coefficient of regression (9.51624), the highest t-value (4.61) and the lowest *p*-value (<0.001) given by the ANOVA test, we can consider that there exists significant linear regression, which allows us to conclude that there is a statistically significant relationship between the predictor variable and the response variable. Furthermore, the equation showed significant interaction between factors with positive coefficients. The interaction between SC and temperature (X_1,2_) positively impacted the results, illustrated as an example of experiences 3 and 14 (Table 3) for 5 h of extraction. Additionally, the interaction between SC and time (X_1,3_) improved the results, as shown in experiences 13 and 14 (Table 3).

As stated by the Pareto of proteins, temperature is the most effective factor in the response, with an η^2^ = 41.21% (Figure 4). However, time and SC concentration are not effective factors. The F-test value for protein response has been improved and is higher than F_9-5_ -tab (Table 4), which allows the validation of the second model for this response.

The best extraction of proteins was achieved in this model, represented by the surface plot showing that it exists in the optimal area, with the maximum temperature (100 °C), time (5 h), and SC concentration around 7.5%.

### 2.8. Effects of Extraction Variables on Carbohydrates

Statistical analysis of both optimization models studied in this work demonstrates that the second model is fitted for carbohydrate response. The F-test value has been ameliorated in the second model with a higher value than F_9-5_ -tab (10.01) (Table 4), which allows the validation of the studied model for this response. The Pareto chart shows temperature η^2^ = 40.81% is the only compelling factor (Figure 5).

As we reached a good amount of extraction with the best combination studied (7.5% SC, 100 °C, and 5 h), we can consider this as the best extraction for this study, with an amount around 65 mg·g^−1^ DW.

Analyzing the results with the RSM, the equation for obtaining the optimum conditions for proteins is represented by the linear equation:Carbohydrates = 14.45697 − 1.39821 * X_1_ − 9.51605 * X_1_^2^ + 16.230687 * X_2_ +10.77679 * X_2_^2^ + 3.12962 * X_3_ − 7.05126 * X_3_^2^ + 1.86372 * X_12_ + 0.33342 * X_13_ + 5.59407 * X_23_ (R^2^ = 0.90) 

Statistical parameters such as R^2^ (0.90) and the positive coefficient of regression (14.45697) allow us to conclude a significant linear regression and a statistically significant relationship exists between the predictor variable and the response variable. Also, the equation shows a positive interaction between the three studied factors. As an example of positive interaction, we can mention the case of SC and temperature (X_1,2_) and examples of experiences 3 and 14 (Table 3). Time has positively impacted the carbohydrate content, as shown between experiences 13 and 14 (41.62 and 65.56 µg glucose eq per mg DW) (Table 3).

The best extraction of carbohydrates was achieved in this model, represented by the surface plot showing that it exists in the optimal area, with the maximum temperature (100 °C), time (5 h), and SC concentration around 7.5%.

### 2.9. Effects of Ultrasound Pretreatment

The combination of the ultrasound pretreatment and the best conditions found in alkaline hydrolysis (12.5% SC and 100 °C) was effective regarding the results of polyphenols, antioxidant capacity, proteins, and carbohydrate contents. The results of different extractions are represented in the table (Figure 6).

The multiple comparisons using a post-hoc test (Tukey HSD) have demonstrated a significant difference in polyphenol content and ABTS capacity between the different extractions. Ultrasound pretreatment has shown a low yield in all compounds compared with the other experiences, which can be explained by the importance of temperature in alkaline extraction. However, ultrasound treatment followed by 5 h of alkaline hydrolysis has demonstrated the highest yield of extraction (8.30 ± 0.07 mg GAE per g DW), which exceeded the best yield of alkaline hydrolysis (3.32 ± 0.05 mg GAE per g DW). The maximum antioxidant capacity was obtained with the same treatment (5 h of extraction following the pretreatment) (749.69 µmol TEAC per g DW) (Figure 6).

For protein content, results showed a significant difference between different extractions except for 3 h and 5 h of hydrolysis, and the alkaline hydrolysis only (23.64 ± 0.1, 24.39 ± 0.19, and 23.58 ± 0.21 mg BSA eq per g DW, respectively) was the best extraction for this compound. Concerning carbohydrates, the highest yield was achieved with 5 h of hydrolysis followed by the pretreatment and the alkaline hydrolysis only (60.54 ± 0.28 and 62.98 ± 0.67 mg glucose eq per g DW). Therefore, applying UAE before the alkaline hydrolysis did not significantly improve proteins and carbohydrates compared to only 8 h of alkaline hydrolysis.

## 3. Discussion

The analysis of the *C. cylindracea* composition is crucial for evaluating its nutritional value and bioactive potential for application in the food, pharmaceutical, and cosmeceutical industries. Ash content represents the inorganic fraction of the biomass after the combustion. It gives an insight into the mineral content, which is important for evaluating an organism’s nutritional and health benefits [32]. Ash content was evaluated in different *Caulerpa* species with a variance in the results depending on the sample; as an example, in the study of Kasmiati et al., *C. lentilifera* exhibited a higher mineral content than *C. racemosa* (63.83% and 34.44%, respectively) [33]. In the present study, the ash content ranged from 17.16% to 26.51%, with the lowest amount for the winter. Seasonal variation was illustrated in the study of the brown seaweed *Sargassum wightii*, which showed a higher ash content in the dry season with a maximum content in July (22.30 ± 0.12%) [34], which may justify the low values found in the present study.

Lipids are a valuable compound in macroalgae that is gaining interest in scientific research for its interesting biological properties, such as antioxidant, antimicrobial, anti-inflammatory, and anti-proliferative [35]. These bioactive potentials can be highlighted in different applications, such as nutricosmetic and pharmaceutical [36,37]. In this study, the total lipid content was comparable between the various samples. The amount of lipids ranges from 2.23% to 2.83%, and they are similar to lipid content from different *Caulerpa* sp. from the Indian coast (2.80%, 3.06%, and 2.64% for *C. veravelensis*, *C. scalpelliformis*, and *C. racemosa*, respectively) [34]. On the contrary, the results are lower than those reported in *C. lentillifera* from the Hawaiian coast, which presented around 7.00% of lipids [38].

The FA composition of the three samples was determined in the present study, showing the presence of different fatty acid groups. Results showed that *C. cylindracea* contains essential FA, Omega-3, and Omega-6 PUFA, such as linoleic acid (LA) (C18:2ω6) and α-linolenic acid (ALA) (C18:3⍵3). In the study of Matanjun et al., *C. lentillifera* showed that the most abundant FA is palmitic acid (33.78%), comparable to the results of *C. cylindracea* (around 32.00%), followed by oleic acid ω9 (C18:1) (32.49%), which is higher than the results shown in this study, which were around 10% for CSS1 and CWS1 and 8.5% for CSS2. Moreover, the ratio ω6/ω3 (1.07) was similar to the ratio found for CSS1 and CWS1 (1.09 and 1.49, respectively) but slightly lower than CSS2 (1.49) [39]. In comparison to our study, different *Caulerpa* species from the study of Nagappan et al. showed the presence of various FA compositions. It can be remarked that palmitic acid was lower than this reported in *C. cylindracea* (8.47%, 10.82%, and 12.65% for *C. lentillifera*, *C. racemosa* var. *clavifera*, and *C. racemosa* var. *laeteviens*, respectively). The essential PUFAs found in this study, such as (ω6) linoleic acid (C18:2) and (ω3) α-linolenic acid (C18:3), were identified in different *Caulerpa* species; linoleic acid C18:2 was around 4.00%, comparable to *C. cylindracea*. However, α-linolenic acid (C18:3) was slightly higher in this study, particularly for CSS1 and CWS1, which exhibited around 6% [40]. In another study, *C. macrodisca* showed almost all the different FAs spotted in this work. Palmitic acid was the most abundant FA (17.29%), but lower than the result of this study. Oleic acid was higher than the findings in this study (14.18%). Total PUFA detected in *C. macrodisca* was higher than in *C. cylindracea* samples, and the ratio (ω6/ω3) is almost double for *C. macrodisca* [41]. The ratio ω6/ω3 in C. cylindraceae ranged from 0.96 to 1.49. The proportion between 1.0 and 2.0 is considered very healthy [42]. This balanced ratio of ω-6 to ω-3 is critical to human development during pregnancy and lactation, preventing and managing chronic diseases [43,44].

Proteins from seaweed are capturing a vital interest due to their different possible utilizations in pharmaceutical, biomedical, and nutritional applications. Depending on the phylum, protein content can range from 9% to 47% DW [45]. According to the study of Marquez et al. [46], the protein amount in the *Caulerpa* genus can significantly vary between species (from 3.98% to 18.3%). Our findings highlighted protein content from different samples of *C. cylindracea* with no significant differences considering the site and the season. The total proteins in *C. cylindracea* are lower than those found in different *Caulerpa* species [40,47]. However, the results are comparable to those found in *C. veravelensis* (7.77% DW) and *C*. *scalpelliformis* (10.05% DW). The carbon-to-nitrogen ratio (C:N) indicates nutritional characteristics. In this study, the C:N ratio was slightly higher in CWS1 (22.00) than in summer samples from both sites (17.37 and 16.97 for CSS1 and CSS2, respectively). The results can be compared to those reported in other *Caulerpa* species in the study of Kumar et al. where the ratios are 17.57, 16.33, and 18.45 for *C. veravelensis*, *C. scalpelliformis*, and *C. racemosa*, respectively [34].

The lack of significant differences in total lipids and proteins between the samples of *C.cylindracea* harvested from two sites and two seasons (summer and winter) can be explained through different viewpoints. The similarity between the results of samples from other sites can be interpreted by a similar concentration of dissolved nutrients such as nitrogen and phosphorus, which can affect the growth of the algae [48]. It could be suggested that abiotic factors like temperature, salinity, light intensity, pH, and nutrient composition that can influence the composition of the algae are not different between the two sites [49].

Previous studies showed a seasonal variation in primary and secondary metabolites from different algae [50]. However, this study found no meaningful distinction between the two seasonal samples. For total lipids, CWS1 showed a slightly higher content of 2.86%, and the composition of FA showed almost the same. For proteins, results were comparable, and for %C and %H, CSS1 showed lower content (28.34%) in comparison with CWS1 (35.02%), but for sulfur, CSS1 showed a higher concentration (2.65%). This could be explained by the mild winter and the seasonal change that may not inflict sufficient differences in this area. Further studies must be conducted to investigate the composition of different seasonal samples.

Morphologically, *C. cylindracea* and other Caulerpacea species (Bryopsidales) exhibit a unique cellular organization. Subsequently, its cell wall composition presents a fibrillar structure with a complex sulfated polysaccharide, reinforcing and stabilizing the cell wall framework [51,52]. It may contain different monosaccharides such as galactose, glucose, mannose, xylose, and arabinose [53]. In addition, polysaccharide is a primary element that plays an essential role in protection against environmental stress and adhesion and cell interaction [54,55,56]. Several techniques are currently employed for cell wall disruption, enhancing the extraction of these valuable compounds. These methods can be mechanical/physical, such as sonication, microwave, or pressure-assisted methods, or chemical, including acid hydrolysis, alkaline hydrolysis, or enzymatic treatment [57,58,59]. In the present study, we used an alkaline hydrolysis; hence, we optimized to find the best concentration of SC, temperature, and extraction time. This method uses SC at elevated temperatures to disrupt cell walls and release intracellular compounds. Compared to water extraction, alkaline hydrolysis, as a simple and cost-effective method, can enhance the extraction of soluble compounds. However, this approach efficiently isolates different compounds that may differ from one species to another, notably for target compounds [60]. Enhancing extraction techniques from invasive species can contribute to sustainable resource management, turning ecological challenges into economic opportunities. Additionally, the quest for more efficient extraction methods for bioactive compounds with considerable economic value improves extraction efficiency while simultaneously optimizing the isolation of different compounds.

Polyphenols are polar compounds containing aromatic rings and hydroxyl groups. Thus, its extraction uses polar protic solvents such as ethanol, methanol, and water and polar aprotic solvents, for instance, ethyl acetate and acetone [61,62]. Several polyphenols have been identified in *Caulerpa* sp., including various flavonoids such as catechin and epicatechin and different phenolic acids like gallic acid, ellagic acid, chlorogenic acid, and hydroxybenzoic acid, and some species may contain bromophenols [23,63]. For polyphenols, the best extraction was achieved using 12.5% SC at 100 °C for 8 h with an amount of around 3.33 mg GAE·g^−1^ DW, and this shows that increasing SC concentration negatively impacts the polyphenol content. The present study showed a higher content of polyphenols compared with other studies on *C. racemosa,* which exhibited a content ranging from 0.010 to 0.013 mg GAE·g^−1^ DW using 70% methanol with ultrasound-assisted extraction, 67.89  ±  3.88 mg·100 g^−1^ dry sample using 60% ethanol and microwave, and a phenolic content of 144 ± 22 mg GAE·100 g^−1^ using 50% methanol [64,65,66]. The solvent also impacts the final content, as shown in the *C. racemosa* and *C. lentillifera* study, where chloroform extract exhibited the best content for both algae compared with methanol and water [13]. Despite that, we employed water extraction in this study to prevent the toxicity associated with organic solvents, ensuring biosecurity for food or cosmeceutical applications.

The drying method can also influence the phenolic content, which was studied in the work on *C. lentillifera,* where freeze-drying showed a higher content of 2.04 ± 0.03 mg GAE·g^−1^ DW compared to the thermal drying with 1.30 ± 0.02 mg GAE·g^−1^ [67]. Comparing our results to previous studies, it can be noted that polyphenol content depends on the extraction method, as mentioned in comparing the phenolic content in *C. racemosa* collected from Indonesia between two methods using hot water extraction and subcritical water extraction [68]. Different *Caulerpa* species exhibited a higher polyphenol content than this study, and the extraction was conducted using an ultrasonic water bath for 1 h with methanol [69]. In contrast, our results are higher than the phenolic content found in *C. racemosa* (1.44 ± 0.22 mg GAE·g^−1^) using 50% methanol and shaking for 1 h [64]. The different methods and solvents could explain these differences. While methanol extraction can exhibit high polyphenols and flavonoids, its usage is not an ideal option for food or cosmetics due to its toxicity.

A positive correlation (r = 0.8712, *p* = 0.000023, *n* = 15) was observed between polyphenols and ABTS, and it was significant. The best result of ABTS (518.70 ± 1.92 µmol TEAC·g^−1^ DW) was found at the same best conditions for polyphenols (12.5% SC at 100 °C for 8 h). Similarly, different studies reported this strong correlation between the antioxidant capacity and polyphenol content. For instance, the values of EC50 determined by the radical scavenging test DPPH of four *Ulva* species are correlated to the total phenolic content [70]. Additionally, the study of the antioxidant activity of different green seaweeds highlighted a significant correlation between polyphenol content and radical scavenging activity DPPH [71]. All these results corroborate the information that phenolics are one of the main antioxidant compounds in algae and plants [72,73,74].

Total protein content has reached 25.33 ± 0.23 mg·g^−1^ in alkaline hydrolysis using 7.5% SC, 100 °C, and 5 h. The optimization results showed that the increase in SC to 15% negatively impacted the final contents, and the best temperature for extraction was 100 °C. Published work has shown notable variations in protein yield across different protocols. The extraction with deionized water and Tris HCl buffer yielded less than trichloroacetic acid (TCA) extraction buffer and Tri Reagent (TRE) methods. Furthermore, a difference in yield was noted between the species of *Caulerpa* species; for example, TCA extraction (first protocol at 4 °C) exhibited higher amounts for *C. prolifera* and *C. cylindracea* than *C. taxifolia* [75]. In the present study, it was proved that alkaline hydrolysis combined with temperature enhances soluble protein content. Exciting literature established that different *Caulerpa* species are promising sources for nutritional utilization with a significant amount of proteins [76].

Total carbohydrates showed high content (65.56 ± 0.20 mg·g^−1^ DW) when performing an extraction with 7.5% SC at 100 °C for 5 h. Previous studies showed that *Caulerpa* sp. contains a considerable amount of total carbohydrates, as the investigation of the chemical composition of *C. lentilifera* revealed a high amount of 6.5% soluble fiber [77]. Also, a study about the nutritional value of the different edible species of Caulerpa has shown that carbohydrate content ranges from 40% to 52% [40]. Previous studies have shown that alkaline hydrolysis is an effective method for extracting insoluble polysaccharides. Indeed, expanding the cell wall interferes with the hydrogen bonds that link cellulose and hemicellulose. As a result, the hemicellulose could be solubilized and converted into a soluble carbohydrate [78].

Ultrasound-assisted extraction is one of the novel methods that can be used to increase the extraction yield and the recovery of bioactive compounds. Therefore, a combination of ultrasound pretreatment was conducted to analyze the enhancement of the contents. As a result, a significant improvement was noticed, particularly with polyphenols and ABTS. Recent studies showed an improvement in polysaccharide yields from macroalgae [79]. A pretreatment of UAE with a solid-liquid extraction enhanced the yield of *Ulva lactuca* polysaccharides, reaching 17.57% using alkaline extraction (2% NaOH at 90 °C for 5 h) combined with 1 h of UAE [29]. Similarly, the positive effect of the combination of UAE with the conventional method was observed in the alginate yields from *Sargassum muticum* (13.6%) [80]. In addition, the UAE was allowed to reach more carrageenans from *Kappaphycus alvarezii* and *Euchema aenticulatum* (50–55%) compared to the conventional method, which exhibited only 27% [31]. Another study on the investigation of different parameters utilizing UAE has proven the enhancement of extraction yield of polyphenols and phlorotannins from *Fucus vesiculosus* (704.9 mg·g^−1^ and 568.9 mg·g^−1^ for polyphenols and phlorotannins, respectively) [81]. Several studies explored the efficiency of ultrasound extraction on polyphenol content by optimizing different factors, including solvent-to-material ratio, temperature, time, and ultrasound power [82]. Moreover, due to its high versatility, UAE has also been used in phycobiliprotein extraction using maceration in combination with UAE from the red algae *Gelidium pusillum* has proven an enhancement of extraction from 58.77% to 76.80% and from 63.2% to 93.13% for R-PE (Phycocyanin-Phycoerythrin) and R-PC (Phycocyanin-Phycoerythrin), respectively [83].

## 4. Materials and Methods

### 4.1. Biological Material

The green thallophytes of *C. cylindraceae* were collected from two different sites. The first site is Kuriat Island, Tunisia (Site 1, 35°45′ N, 11° 00′ E), in different seasons: winter “CWS1” (February 2024) (temperature 15 °C; pH 8.13; salinity of 37 psu; 0.03 mmol·m^−3^ of NO_3_^−^; 0.04 mmol·m^−3^ of PO_4_^−^; 243.4 mmol·m^−3^ of O_2_ (‘My Ocean Pro—Copernicus Marine Open Data’ platform)) and summer “CSS1” (June 2024) (temperature 23 °C; pH 8; salinity of 37.93 psu; 0.016 mmol·m^−3^ of NO_3_^−^; 0.026 mmol·m^−3^ of PO_4_^−^; 225.5 mmol·m^−3^ of O_2_ (‘My Ocean Pro—Copernicus Marine Open Data’ platform)), and the second site is Monastir coast, Tunisia (Port of the Skanes Presidential Palace Habib Bourguiba) (Site 2, 35°46′ N, 10° 46′ E) in summer “CSS2” (June 2024). (temperature 23 °C; pH 8; salinity of 38 psu; 0.014 mmol·m^−3^ of NO_3_^−^; 0.031 mmol·m^−3^ of PO_4_^−^; 221 mmol·m^−3^ of O_2_ (‘My Ocean Pro—Copernicus Marine Open Data’ platform)). Then, the algae were transported to the research laboratory at the Higher Institute of Biotechnology of Monastir, Tunisia, in plastic containers filled with seawater. Then, the thalli were washed with tap water and oven-dried at 35 °C for 2 days, then ground and stored in the freezer (−20 °C) until analysis.

### 4.2. General Characterization Prior to Alkaline Hydrolysis Optimization

#### 4.2.1. Ash Content

The weight of inorganic compounds was determined according to Ismail’s method [32]. In brief, 1 g of dry material of *C. cylindracea* was placed into a weighed crucible and was put in a muffle for 3 h at 550 °C. Throughout this procedure, all organic matter was eliminated. Subsequently, the weight of the sample was calculated by subtracting the final weight from the initial weight. Ultimately, the percentage of inorganic compounds in the sample was determined according to the following formula:Ash content = ((Ca − Ce)/DM) * 100 
where

Ca: Weight of crucible containing ash (g)

Ce: Weight of empty crucible (g)

DM: Weight of dried material (g)

#### 4.2.2. Total Lipids

Lipids were extracted from seaweed using the Folch method [84]. A 200 mg sample of dried seaweed was homogenized with 5 mL of chloroform:methanol (2:1 *v*/*v*) containing 0.01% butylhydroxytoluene (BHT) as an antioxidant. To facilitate phase separation, 2 mL of 0.88% potassium chloride (KCl) solution was added, and the mixture was centrifuged at 1200× *g* for 5 min. This process resulted in two distinct phases: the upper phase was discarded, while the lower lipid-rich phase was collected, filtered, and cooled at −20 °C for at least 20 min. Any thin salt layer formed on the surface was carefully removed using a micropipette. The extracted lipids were weighed and stored under a nitrogen atmosphere to prevent oxidation. The lipid content was calculated as a percentage of the dried biomass weight using the formula:Lipids (%) = [Lipids (g)/Biomass (g DW)] × 100 

#### 4.2.3. Gas Chromatography of Fatty Acids

For the analysis, 50 µL of each sample was derivatized using a methanol:acetyl chloride mixture (20:1 *v*/*v*) to produce fatty acid methyl esters (FAMEs). Briefly, 70 to 90 mg of each sample was placed in a 3 mL derivatization vial, then 1 mL of hexane and 1 mL of methylation reagent (methanol: acetyl chloride, 20:9 *v*/*v*) were added. Subsequently, the mixture was heated at 105 °C for 20 min, then cooled to room temperature. Afterwards, the reaction was stopped by adding 1 mL of Milli-Q water, and the mixture was centrifuged at 2000 rpm for 2 min. Finally, the supernatant (hexane phase) was collected and transferred to an amber glass vial. The analysis used a Trace 1300 Gas Chromatograph (Thermo Scientific^TM^, Waltham, MA, USA) with a Triplus RSH autosampler and a flame ionization detector (FID). Separation was achieved using a CP-Sil 88 capillary column (100 m × 0.25 mm × 0.2 µm, Agilent, Santa Clara, USA) under the following conditions: 1 µL injection in splitless mode, carrier gas flow at 1 mL·min^−1^, and an injector temperature of 270 °C. The oven program started at 140 °C (held for 5 min), increased at 4 °C to 240 °C, and remained constant for 20 min.

The GC-FID operated at 280 °C with airflow at 350 mL·min^−1^, hydrogen at 35 mL·min^−1^, and makeup gas at 40 mL·min^−1^. This setup enabled efficient separation and accurate quantification of FA, ensuring high sensitivity and reproducibility.

The identification of fatty acids in the samples was carried out by comparing the retention time and mass spectra of fatty acids standards previously analyzed under identical conditions.

#### 4.2.4. Elemental Analysis

The total combustion method was used for the determination of total carbon (C), hydrogen (H), nitrogen (N), and sulfur (S) from dried material of *C. cylindracea*, utilizing a LECO TruSpec Micro CHNSO-Elemental Analyzer in the Research Support Central Services (SCAI, University of Malaga, Malaga, Spain). The sample was subjected to complete and instantaneous oxidation via pure combustion with controlled oxygen at a temperature reaching 1050 °C for C, H, N, and S. For O, pyrolysis was carried out at 1300 °C, converting O to CO, followed by oxidation to CO_2_. Thereafter, the final combustion products (CO_2_, H_2_O, SO_2,_ and N_2_) were measured by a selective infrared (IR) absorption detector for C, H, and S. For nitrogen quantification, a differential thermal conductivity sensor (TCD) was used. Results were presented in percentages in terms of the weight of the dry sample for each element (C, H, N, S).

#### 4.2.5. Proteins Content

The protein content was determined by multiplying the percentage of nitrogen (%N) mentioned above by the conversion factor (N-Prot) according to the following equation:Protein content = N% × N-Prot 

The factor used for *Caulerpa* sp. is 5.13, as stated in the study of Lourenço et al. for green macroalgae [85].

### 4.3. Conventional Extraction

Conventional extraction was conducted according to the method of Hossain et al. [86] with some modifications. The dried sample (0.5 g) was extracted with water at a ratio of solvent to material (50:1 mL·g^−1^) at 30 °C. The samples were shaken for 12 h using a shaking water bath. The extracts were then immediately cooled on ice to room temperature, filtered, and diluted to the required volume for analysis.

### 4.4. Alkaline Hydrolysis Extraction

Bioactive compounds were extracted by alkaline hydrolysis in an Eppendorf tube using a thermoblock at the Institute of Blue Biotechnology and Development (IBYDA) of Malaga University (Spain). For each experiment, 30 mg of dried sample were mixed with 1.5 mL of extraction solvent with different concentrations of SC ranging from 0 to 12.5%. The resulting mixture was treated with temperatures ranging from 30 to 100 °C and extraction times from 1 to 8 h. The mixture was vortexed every 30 min. Subsequently, the samples were centrifuged at 10,000× *g*, 4 °C for 10 min; the supernatant was recovered, and the pH was adjusted to 7 using lactic acid. Each experiment was conducted in triplicate.

#### Response Surface Model Design

For this part of the study, the sample used for optimization was the CSS2. The extraction of soluble bioactive compounds using alkaline hydrolysis is influenced by different factors. The present work examined SC concentration, temperature, and extraction time to assess their impact on bioactive compound content and identify the optimal conditions to achieve the best extraction. An optimization experiment using RSM, with three factors and three levels, Central Composite Design (CCD) with 15 experimental runs. Table 5 represents the independent variables, denoted with both coded and uncoded levels, as follows: SC concentration (X_1_, ranging from 0 to 25%), extraction temperature (X_2_, ranging from 30 to 100 °C), and extraction time (X_3_, ranging from 1 to 8 h).

To improve the extraction of proteins and carbohydrates to achieve a significant F-test value, we developed another experimental matrix, as shown in Table 6, modifying levels of independent variables. New independent variables are denoted with both coded and uncoded levels of SC concentration (X_1_, ranging from 0 to 15%) and extraction time (X_3_, ranging from 1 to 5 h), and keeping the extraction temperature the same (X_2_, ranging from 30 to 100 °C) due to the impossibility of increasing the temperature.

The experimental data were analyzed using a second-order polynomial model to determine the regression coefficients (β). The response surface analysis employs the following generalized second-order polynomial model:$$Y=\beta 0+\sum_{i=1}^{k}\beta iXi+\sum_{i=1}^{k}\beta iiXi2+\;\sum_{i\neq j=1}^{k}\beta ijXiXj$$

Y represents the response variable, while Xi and Xj are the independent variables, with k denoting the number of variables tested (k = 3). The regression coefficients are defined as follows: β0 for the intercept, βi for the linear terms, βii for the quadratic terms, and βij for the interaction terms.

ANOVA was conducted at a 95% confidence level to evaluate the significance of the individual linear, quadratic, and interaction regression coefficients using Statistica 10^®^ software (StatSoft, Tulsa, OK, USA). The coefficient of determination (R^2^) was used to assess the fit of the polynomial equation to the response data. Statistical analysis of the significance of all terms in the polynomial equation was carried out by calculating the F-value at a significance level of *p* < 0.05. The software also generated 3D response surface graphs.

### 4.5. Combination of Ultrasound Extraction (UAE) and Alkaline Hydrolysis

UAE was performed as a pretreatment in combination with the best factors of SC and temperature (12.5% SC and 100 °C) from alkaline hydrolysis to enhance the extraction yields and reduce the extraction time. An ultrasonic cell disruptor of 25 kHz was used (BIOBASE UCD-150, 150 W/220 V, 50 Hz) utilizing a 6 mm diameter probe. The extraction was performed using the same solvent-to-material ratio, applying a pulsed cycle (1 s on, 1 s off) and a power ratio of 70% for 60 min at ambient temperature. Subsequently, the mixture was set up for hot extraction (100 °C) for different hours: 0 h, 1 h, 3 h, and 5 h. Afterward, samples were left to cool and centrifuged at 10,000× *g*, 4 °C for 10 min.

### 4.6. Quantification of Bioactive Compounds

#### 4.6.1. Polyphenols

Phenolic compound analysis was conducted using the spectrophotometric Folin-Ciocalteu method [87] with some modifications. Briefly, 100 µL from each extract was mixed with 700 µL of distilled water and 50 µL of Folin-Ciocalteu phenol reagent (Sigma-Aldrich, St. Louis, MO, USA). The mixture was vortexed, 150 µL of Na_2_CO_3_ 20% was added, and incubated in darkness for 1 h at 4 °C. For quantification, a standard curve of gallic acid (GA) was used (0 to 100 µg·mL^−1^). The absorbance was determined using a UV-visible spectrophotometer at 760 nm. All the analyses were performed in triplicate, and the results were expressed as mg of GA per g dry weight (DW).

#### 4.6.2. Antioxidant Capacity

The antioxidant capacity of different extracts was determined by the 2,2′-Azino-bis (3-ethylbenzothiazoline-6-sulfonic acid) (ABTS) radical scavenging assay. The preparation of the radical cation ABTS^+.^ was carried out by mixing 7 mM of ABTS and 2.45 mM of potassium persulfate (K_2_S_2_O_8_) in a sodium phosphate buffer solution (0.1 M, pH 6.5). The mixture was incubated for 16 h in darkness at 4 °C to facilitate the radical formation. At first, the ABTS^+.^ absorbance was adjusted by diluting the mixture in phosphate buffer until an absorbance level at 727 nm was 0.75 ± 0.05. For the reaction, the assay was conducted per the method [88], mixing 950 µL of the diluted ABTS^+.^ with 50 µL of different extracts. The mixtures were incubated for 8 min in obscurity, and then the absorbance (DOe) was measured using a UV-visible spectrophotometer at 727 nm.

The initial absorbance (DOi) (blank) was measured, and the percentage of antioxidant activity (AA%) was calculated according to the following formula:AA% = [(DOi − DOe)/DOi] × 100 

The antioxidant compound concentrations were assessed using a Trolox standard curve (6-hydroxy-2,5,7,8-tetramethylchroman-2-carboxylic acid) (Sigma-Aldrich), which ranged from 20 to 100 µg·mL^−1^. Results were expressed as µmol of eq Trolox (TEAC) per g DW.

#### 4.6.3. Soluble Proteins

Soluble protein analysis used the spectrophotometric Bradford method [89]. For the reaction, 750 µL of phosphate buffer (0.1 M, pH 6.5) and 200 µL of the Bradford reagent (BioRad, Fort Worth, TX, USA) were added to 50 µL of each sample. The mixture was vortexed and incubated for 15 min in room-temperature darkness. The absorbance was measured at 595 nm using a UV-visible spectrophotometer. A standard curve of different concentrations of bovine serum albumin (BSA) (Sigma-Aldrich, Steinheim, Germany) ranging from 4 to 60 µg·mL^−1^ was used for the quantification. All the analyses were carried out in triplicate, and the results were expressed in mg BSA per g DW.

#### 4.6.4. Soluble Carbohydrates

Soluble carbohydrate analysis was carried out using the spectrophotometric method [90]. Briefly, 1 mL of 0.2% anthrone in sulfuric acid (*w*/*v*) was added to 500 µL of each extract, and the mixture was incubated at 100 °C for 3 min in the dry bath. After being cooled, the absorbance was determined at 630 nm using a UV-visible spectrophotometer. The quantifications were determined by a standard glucose curve of different concentrations ranging from 25 to 200 µg·mL^−1^. The analysis was carried out in triplicate, and the results were expressed in mg of glucose per g DW.

Figure 7 provides a summary of the entire methodology used in the present study to enhance understanding.

### 4.7. Statistical Analysis

A Design of Experiments (DOE) approach using Box-Behnken designs was implemented to assess the impact of three independent factors (SC concentration, temperature, and time) on the extraction of different soluble compounds with alkaline hydrolysis. The experiments were implemented through an RSM using Statistica 10^®^ software (StatSoft, Tulsa, OK, USA).

Statistical analysis for comparison of different sample contents was performed using independent samples T-tests (*p* ≤ 0.05) or one-way ANOVA followed by Tukey’s test (*p* ≤ 0.05) for multiple comparisons.

## 5. Conclusions

Considering the findings of this study, *C. cylindracea* emerges as a significant source of bioactive compounds with potential applications. These results provide a promising foundation for exploring the diverse biological properties of this macroalga. Alkaline hydrolysis notably enhances the extraction yield of bioactive compounds, highlighting the importance of further optimization. Additionally, ultrasound pretreatment has demonstrated its effectiveness in increasing yields while reducing extraction time, making it a valuable technique for improving efficiency. The bioactive compounds identified in this study are known to present significant potential, particularly for their antioxidant, anti-aging, anti-inflammatory, antibacterial, and immunomodulatory properties, and these attributes make them highly relevant for the cosmeceutical and nutraceutical industries. Moving forward, further research is essential to maximize the commercial potential of this species while ensuring environmental sustainability. Moreover, these findings underscore the role of ultrasound-assisted extraction in enhancing bioactive compound yields by improving solvent penetration and optimizing extraction efficiency.

## Figures and Tables

**Figure 1 marinedrugs-23-00208-f001:**
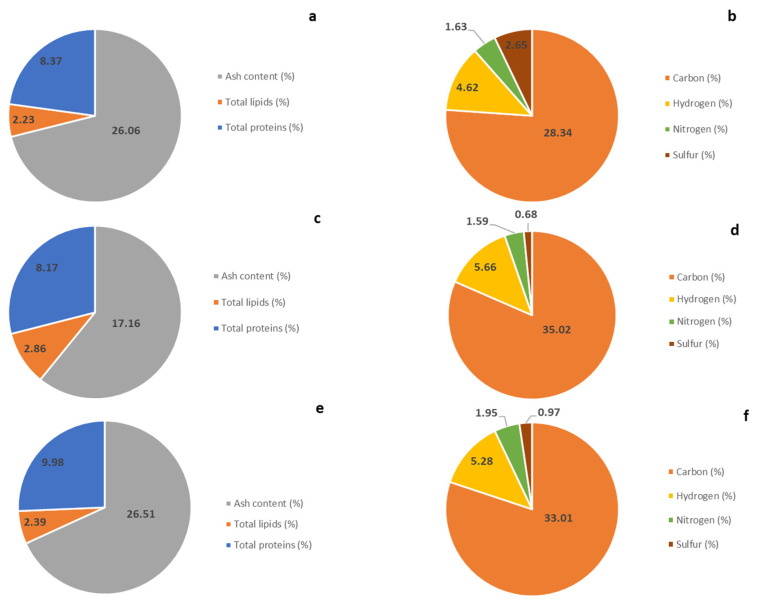
Composition in percentage (%) of ash content, total lipids, total proteins, and organic elements (carbon (C), nitrogen (N), hydrogen (H), and sulfur (S)) of *Caulerpa cylindracea* samples, (**a**,**b**) considering CSS1: *Caulerpa* Summer Site 1, (**c**,**d**) considering CWS1: *Caulerpa* Winter Site 1, and (**e**,**f**) considering CSS2: *Caulerpa* Summer Site 2. The discrepancy between the total sum of all measured parameters and 100% can be explained by the standard deviation associated with each measurement, which is not shown in the graphs.

**Figure 2 marinedrugs-23-00208-f002:**
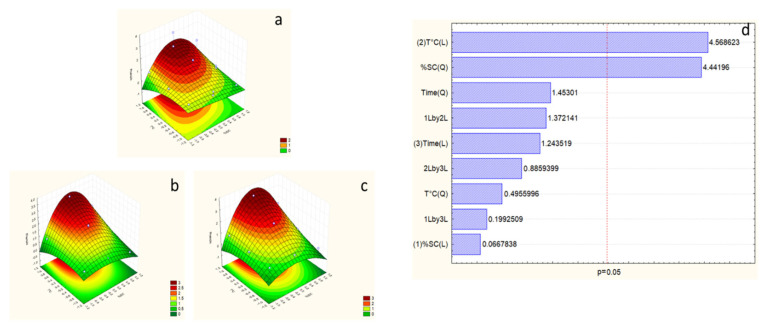
Surface plot ((**a**) considering sodium carbonate (SC) and temperature at time = −1 (1 h), (**b**) considering SC and temperature at time = 0 (4.5 h), and (**c**) considering SC and temperature at time = 1 (8 h)), and Pareto chart (**d**) of polyphenol response for the first optimization experiment. The numbers (1), (2), and (3) in Pareto chart refer to the order of the factors in the statistical analysis and do not affect the interpretation of the results.

**Figure 3 marinedrugs-23-00208-f003:**
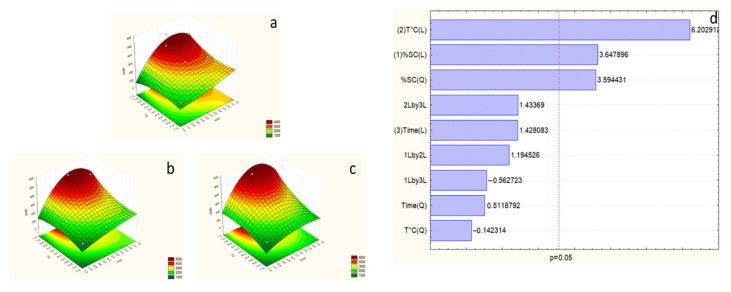
Surface plot ((**a**) considering SC and temperature at time = −1 (1 h), (**b**) considering SC and temperature at time = 0 (4.5 h), and (**c**) considering SC and temperature at time = 1 (8 h)), and Pareto chart (**d**) of antioxidant activity response for the first optimization experiment. The numbers (1), (2), and (3) in Pareto chart refer to the order of the factors in the statistical analysis and do not affect the interpretation of the results.

**Figure 4 marinedrugs-23-00208-f004:**
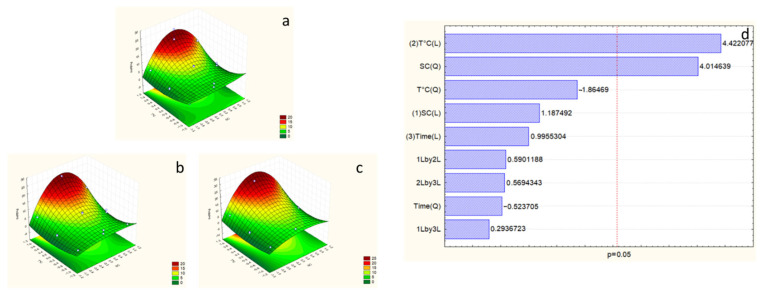
Surface plot ((**a**) considering SC and temperature at time = −1 (1 h), (**b**) considering SC and temperature at time = 0 (3 h), and (**c**) considering SC and temperature at time = 1 (5 h)), and Pareto chart (**d**) of protein response for the second optimization experiment. The numbers (1), (2), and (3) in Pareto chart refer to the order of the factors in the statistical analysis and do not affect the interpretation of the results.

**Figure 5 marinedrugs-23-00208-f005:**
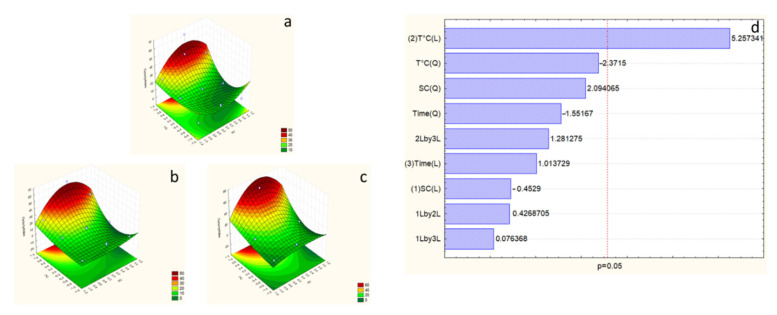
Surface plot ((**a**) considering SC and temperature at time = −1 (1 h), (**b**) considering SC and temperature at time = 0 (3 h), and (**c**) considering SC and temperature at time = 1 (5 h)), and Pareto chart (**d**) of carbohydrate response for the second optimization experiment. The numbers (1), (2), and (3) in Pareto chart refer to the order of the factors in the statistical analysis and do not affect the interpretation of the results.

**Figure 6 marinedrugs-23-00208-f006:**
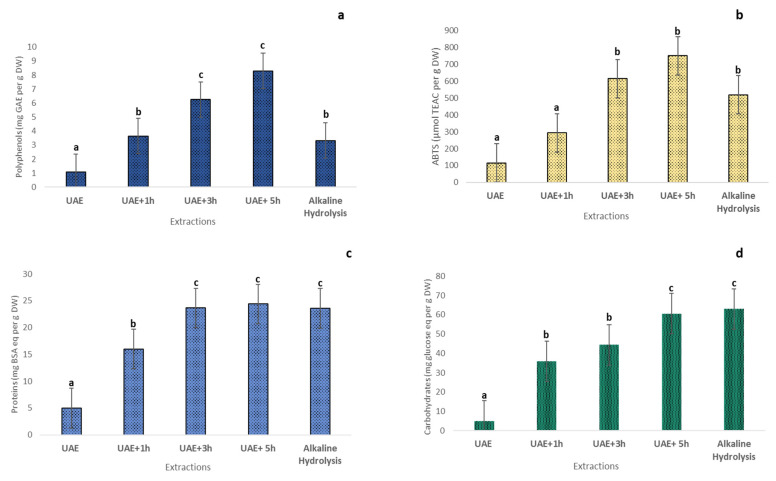
Bioactive compound contents ((**a**) considering polyphenols (mg GAE per g DW), (**b**) considering antioxidant activity ABTS (µmol TEAC per g DW), (**c**) considering proteins (mg BSA eq per g DW), and (**d**) considering carbohydrates (mg glucose eq per g DW) of different extractions using ultrasound-assisted extraction (UAE), comparing with the best response in alkaline hydrolysis (12.5% SC, 100 °C, and 8 h). Different letters indicate significant differences according to the one-way analysis of variance and Tukey’s test (n = 3, *p* ≤ 0.05).

**Figure 7 marinedrugs-23-00208-f007:**
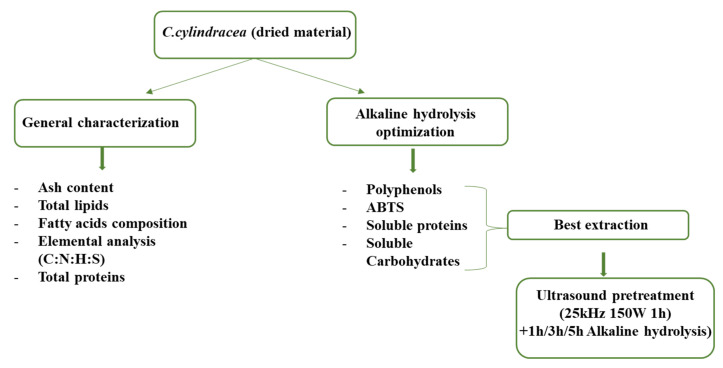
A summary of the entire methodology used in the present study.

**Table 1 marinedrugs-23-00208-t001:** Fatty acid profiles (retention time (min) and % of area) of *Caulerpa cylindracea* samples (CSS1: Caulerpa Summer Site1, CWS1: Caulerpa Winter Site 1 and CSS2: Caulerpa Summer Site 2).

Fatty Acids	RT (min)	% Area		
		CSS1	CWS1	CSS2
Saturated Fatty acids (SFA)				
C12:0 lauric	21.47	0.47	0.39	0.47
C14:0 myristic	24.90	4.63	4.95	3.67
C16:0 palmitic acid	28.41	33.7	31.87	32.91
C18:0 stearic acid	31.80	1.77	1.50	1.54
C21:0 heneicosanoic	38.55	0.17	0.19	0.17
C21:0 heneicosanoic acid	37.99	0.57	0.44	0.60
C22:0 behenate	39.38	0.24	0.14	0.19
Monounsaturated Fatty acids (MUFA)				
C16:1 palmitoleic acid	29.69	2.71	3.31	2.65
C18:1(ω 9) cis oleic acid	32.92	10.51	8.50	10.19
C22:1 erucate	40.13	0.16	0	0
Polyunsaturated Matty acids (PUFA)				
C18:2 (ω6) linoleic acid	34.58	4.19	4.01	4.37
C18:3 ω6 γ-linolenic acid	35.86	0.48	0.65	0.37
C18:3 ω3 α-linolenic acid	36.54	6.06	6.62	4.37
C20:3 ω6 eicosatrienoic acid	40.39	1.95	1.71	1.80
C21:4 arachidonic acid methyl ester	42.40	0.89	0.84	0.79
C22:2 cis13,16-docisadienoic	43.08	1.43	1.30	1.14
SFA		41.55	39.39	39.64
MUFA		13.38	11.81	12.84
PUFA		10.73	11.28	9.11
PUFA ω6		6.62	6.37	6.54
PUFA ω3		6.06	6.62	4.37
Ratio ω6/ω3		1.09	0.96	1.49

**Table 2 marinedrugs-23-00208-t002:** Central Composite Design (CCD) with responses of the first alkaline hydrolysis extraction of bioactive compounds from *Caulerpa cylindracea* from CSS2. X_1_ refers to sodium carbonate concentration (%), X_2_ to temperature (°C), X3 to time (h), Y_1_ to polyphenols (mg GAE per g DW), Y_2_ to antioxidant activity ABTS (µmol TEAC per g DW), Y_3_ to proteins (mg BSA eq per g DW), and Y_4_ to carbohydrates (mg glucose eq per g DW).

Runs	X_1_	X_2_	X_3_	Y_1_	Y_2_	Y_3_	Y_4_
**1**	0	30	4.5	0.61 ± 0.03	132.35 ± 0.11	2.69 ± 0.05	5.77 ± 0.08
**2**	12.5	30	1	0.30 ± 0.09	190.69 ± 0.55	2.90 ± 0.09	13.63 ± 0.20
**3**	12.5	30	8	0.62 ± 0.02	194.52 ± 0.128	5.41 ± 0.05	12.22 ± 0.16
**4**	25	30	4.5	0.15 ± 0.005	227.17 ± 0.21	3.26 ± 0.002	5.02 ± 0.19
**5**	0	65	1	0.67 ± 0.003	167.14 ± 0.20	2.61 ± 0.019	19.73 ± 0.24
**6**	0	65	8	0.68 ± 0.008	215.45 ± 0.76	3.15 ± 0.03	21.16 ± 0.23
**7**	12.5	65	4.5	2.09 ± 0.01	332.11 ± 0.91	7.16 ± 0.03	21.29 ± 0.12
**8**	12.5	65	4.5	2.08 ± 0.02	330.29 ± 1.25	7.18 ± 0.10	21.03 ± 0.25
**9**	12.5	65	4.5	2.11 ± 0.01	330.25 ± 0.18	7.27 ± 0.04	21.46 ± 0.24
**10**	25	65	1	0.40 ± 0.02	277.83 ± 3.20	1.47 ± 0.09	13.99 ± 0.44
**11**	25	65	8	0.60 ± 0.016	275.05 ± 1.00	0.75 ± 0.05	10.62 ± 0.16
**12**	0	100	4.5	0.84 ± 0.003	217.26 ± 1.03	5.27 ± 0.06	27.83 ± 0.17
**13**	12.5	100	1	2.13 ± 0.07	384.70 ± 1.64	11.69 ± 0.14	21.71 ± 0.07
**14**	12.5	100	8	3.33 ± 0.05	518.70 ± 1.92	23.58 ± 0.21	62.98 ± 0.67
**15**	25	100	4.5	1.73 ± 0.03	420.53 ± 0.83	14.06 ± 0.18	25.51 ± 0.25

**Table 3 marinedrugs-23-00208-t003:** Central Composite Design (CCD) with responses of the second alkaline hydrolysis extraction of bioactive compounds from *Caulerpa cylindracea* from CSS2. X_1_ refers to sodium carbonate concentration (%), X_2_ to temperature (°C), X3 to time (h), Y_1_ to proteins (mg BSA eq per g DW), and Y_2_ to carbohydrates (mg glucose eq per g DW).

Runs	X_1_	X_2_	X_3_	Y_1_	Y_2_
1	0	30	3	1.47 ± 0.05	4.46 ± 0.08
2	7.5	30	1	4.61 ± 0.13	10.20 ± 0.02
3	7.5	30	5	6.94 ± 0.18	11.76 ± 0.26
4	15	30	3	3.59 ± 0.16	4.66 ± 0.07
5	0	65	1	2.32 ± 0.23	17.20 ± 0.25
6	0	65	5	1.94 ± 0.1	16.30 ± 0.23
7	7.5	65	3	9.70 ± 0.23	14.74 ± 0.31
8	7.5	65	3	9.55 ± 0.18	14.67 ± 0.17
9	7.5	65	3	9.30 ± 0.22	13.94 ± 0.43
10	15	65	5	3.05 ± 0.11	7.01 ± 0.22
11	15	65	5	4.77 ± 0.05	7.44 ± 0.23
12	0	100	3	5.34 ± 0.04	23.04 ± 0.34
13	7.5	100	1	18.94 ± 0.06	41.62 ± 0.12
14	7.5	100	5	25.33 ± 0.23	65.56 ± 0.23
15	15	100	3	11.67 ± 0.08	30.70 ± 0.32

**Table 4 marinedrugs-23-00208-t004:** Summarizing ANOVA table data for first and second optimization.

	Responses	Optimization 1	Optimization 2
F-test value F_9,5-_tab = 4.77	Phenols	5.12	-
	ABTS	7.86	-
	Proteins	3.40	4.89
	Carbohydrates	4.72	10.01
R^2^	Phenols	0.90	-
	ABTS	0.93	-
	Proteins	0.86	0.89
	Carbohydrates	0.89	0.90
Related Error of Best extraction	Phenols	15.08	-
	ABTS	8.06	-
	Proteins	14.92	14.03
	Carbohydrates	11.53	12.69

**Table 5 marinedrugs-23-00208-t005:** Coded and uncoded levels of independent variables used in the first RSM design.

Symbols	Independent Variables	Coded Levels		
		−1	0	1
**X_1_**	Sodium Carbonate (%)	0	12.5	25
**X_2_**	Temperature (C)	30	65	100
**X_3_**	Time (h)	1	4.5	8

**Table 6 marinedrugs-23-00208-t006:** Coded and uncoded levels of independent variables used in the second RSM design.

Symbols	Independent Variables	Coded Levels		
		−1	0	1
**X_1_**	Sodium Carbonate (%)	0	7.5	15
**X_2_**	Temperature (C)	30	65	100
**X_3_**	Time (h)	1	3	5

## Data Availability

Data are available upon request.

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
