# Peer review of "Biochemical Composition and Alkaline Extraction Optimization of Soluble Bioactive Compounds from the Green Algae Caulerpa cylindraceae"

_marinedrugs, 2025, doi:10.3390/md23050208_

Round 1
Reviewer 1 Report
Comments and Suggestions for Authors
As we all know, extraction from algae can present serious problems, so the authors detail about application of ultrasound and basic media which has some merrit, but is it necessary to give the numbers of all the contents and concentrations with two decimals, given the known variability of the organisms? What is missing totally in my opinion is if this treatments show any influence on yield and composition of fatty acids, triglycerides and all the lipids, with hinsight of a treatment with base. Were the lipids extracted and analysed before base treatments and compared with a procedure combining base treatment. This things should be detailed and result concerning this should be included. The diagram how to treat algae comes late in the text to late to explain doubts in the readers mind.
Author Response
Reviewer 1 - Round 1
As we all know, extraction from algae can present serious problems, so the authors detail about application of ultrasound and basic media which has some merrit, but is it necessary to give the numbers of all the contents and concentrations with two decimals, given the known variability of the organisms? What is missing totally in my opinion is if this treatments show any influence on yield and composition of fatty acids, triglycerides and all the lipids, with hinsight of a treatment with base. Were the lipids extracted and analysed before base treatments and compared with a procedure combining base treatment. This things should be detailed and result concerning this should be included. The diagram how to treat algae comes late in the text to late to explain doubts in the readers mind.
Thank you very much for taking the time to review this manuscript. Please find the detailed responses below and the corresponding revisions/corrections highlighted/in track changes in the re-submitted files.
Response: We agree, and all the results were presented with two decimals.
Response: Regarding the inquiry about the application of basic media and its effect on lipid/fatty acids extraction, we would like to respectfully clarify that alkaline hydrolysis is not an appropriate method for the extraction of this compounds. Lipids and fatty acids are predominantly non-polar compounds, and their efficient extraction typically relies on the use of organic solvents. On the other hand, alkaline hydrolysis is commonly employed to solubilize polar biocompounds, such as proteins and carbohydrates, and is not effective for the recovery of non-polar lipid fractions. In fact, exposure to strong basic conditions may lead to saponification of fatty acids, thereby complicating the quantification and compositional analysis of intact lipid molecules such as triglycerides. Consequently, in our experimental design, lipid and fatty acids extraction and characterization were conducted independently from the alkaline treatment steps to avoid such degradation. For this reason, we did not include a comparative analysis of lipid/fatty acids profiles before and after base treatment, as it would not yield scientifically meaningful results and could potentially mislead interpretations regarding lipid/fatty acid stability and recovery.
Reviewer 2 Report
Comments and Suggestions for Authors
The main question addressed by the research is the optimization of alkaline extraction conditions for soluble bioactive compounds from the green algae Caulerpa cylindracea using response surface methodology and ultrasound-assisted extraction. The study aims to identify optimal conditions for extracting polyphenols, proteins, carbohydrates, and compounds with antioxidant capacity.
This research adds a systematic study using response surface methodology (RSM) and ultrasound-assisted techniques to optimize the extraction process, which might improve the efficiency and yield over traditional extraction methods.
- There are numerous studies in the literature that explore alkaline hydrolysis extraction using ultrasound-assisted extraction (UAE) and microwave-assisted extraction of algae. The author should reference these studies in the introduction and discuss how this research differs from previous work.
- Tables 1 show the composition of different bioactive compounds. It would be clearer to represent data using histogram figures instead of tables with number for comparison.
- Many data are shown using tables. Author should consider other ways (bar graphs, line graphs or histograms) to display data instead of only tables with numbers.
- For table 2, author could add the gas chromatography overlay for better presenting data.
Author Response
Reviewer 2 – Round 1
The main question addressed by the research is the optimization of alkaline extraction conditions for soluble bioactive compounds from the green algae Caulerpa cylindracea using response surface methodology and ultrasound-assisted extraction. The study aims to identify optimal conditions for extracting polyphenols, proteins, carbohydrates, and compounds with antioxidant capacity.
This research adds a systematic study using response surface methodology (RSM) and ultrasound-assisted techniques to optimize the extraction process, which might improve the efficiency and yield over traditional extraction methods.
Thank you very much for taking the time to review this manuscript. Please find the detailed responses below and the corresponding revisions/corrections highlighted/in track changes in the re-submitted files.
1. There are numerous studies in the literature that explore alkaline hydrolysis extraction using ultrasound-assisted extraction (UAE) and microwave-assisted extraction of algae. The author should reference these studies in the introduction and discuss how this research differs from previous work.
Response: We agree, and we added this information and new references in the text.
2. Tables 1 show the composition of different bioactive compounds. It would be clearer to represent data using histogram figures instead of tables with number for comparison.
Response: We agree, and we change this table in a graphic (Fig. 1).
3. Many data are shown using tables. Author should consider other ways (bar graphs, line graphs or histograms) to display data instead of only tables with numbers.
Response: We agree, and we change the table 1 in figure 1, and table 6 in figure 6.
4. For table 2, author could add the gas chromatography overlay for better presenting data.
Response: We agree, and we added this information as a supplementary material.

Reviewer 3 Report
Comments and Suggestions for Authors
The manuscript "Biochemical composition and alkaline extraction optimization of soluble bioactive compounds from the green algae Caulerpa cylindraceae" explores bioactive compounds of the invasive green macroalgae Caulerpa cylindracea and their potential biotechnological applications. Seaweed samples were collected from two locations and in different seasons of the year. The work uses a fairly large number of different methods of characterization of plant raw materials. Elemental analysis and GC were performed. To optimize the extraction of soluble compounds, the response surface methodology was used.
The results of the work can be useful in the field of biotechnology and food chemistry.
The manuscript is well organized and clearly described the information.
Thus, I think that the manuscript can be published in the Marine Drugs just after minor revisions, taking into account the following comments:
- Derivatization method for GC should be described more.
- The aim of this work should be presented in the Introduction.
- The methodology of applying two optimization stages of the same type does not seem to be well-reasoned. It would be more logical to apply a combination of single-factorial DOE and Central Point DOE. Otherwise, it is necessary to justify the chosen approach more correctly.
Author Response
Reviewer 3 – Round 1
The manuscript "Biochemical composition and alkaline extraction optimization of soluble bioactive compounds from the green algae Caulerpa cylindraceae" explores bioactive compounds of the invasive green macroalgae Caulerpa cylindracea and their potential biotechnological applications. Seaweed samples were collected from two locations and in different seasons of the year. The work uses a fairly large number of different methods of characterization of plant raw materials. Elemental analysis and GC were performed. To optimize the extraction of soluble compounds, the response surface methodology was used. The results of the work can be useful in the field of biotechnology and food chemistry. The manuscript is well organized and clearly described the information. Thus, I think that the manuscript can be published in the Marine Drugs just after minor revisions, taking into account the following comments:
Thank you very much for taking the time to review this manuscript. Please find the detailed responses below and the corresponding revisions/corrections highlighted/in track changes in the re-submitted files.
Derivatization method for GC should be described more.
Response: We agree, and we added more information in the text.
The aim of this work should be presented in the Introduction.
Response: We agree, and we added this information in the text.
The methodology of applying two optimization stages of the same type does not seem to be well-reasoned. It would be more logical to apply a combination of single-factorial DOE and Central Point DOE. Otherwise, it is necessary to justify the chosen approach more correctly.
Response: We appreciate your observation. In our study, we applied two stages of optimization using Response Surface Methodology (RSM) with a Central Composite Design (CCD). This decision was based on the specific goals of each optimization stage and the behavior of the response variables observed during preliminary trials. In the first optimization, we analyzed four responses: phenols, ABTS, proteins, and carbohydrates. While phenols and ABTS showed statistically significant F-test values, proteins and carbohydrates did not. Therefore, to improve the extraction of proteins and carbohydrates and achieve significant F-test values, we developed a second experimental matrix, as shown in Table 6, with adjusted levels of the independent variables. This two-step approach was essential, as each class of compounds exhibited distinct optimal extraction conditions. A single experimental design would not have efficiently optimized all target compounds simultaneously. This explanation is provided in the Materials and Methods section (641-646 lines).

Round 2
Reviewer 1 Report
Comments and Suggestions for Authors
It is quite clear that a base treatment before lipid extraction and analysis makes less sense. What is meant is: we know diglycerides bound to cell walls via sugars which are insoluble. This cannot be extracted by simple solvent extraction and/or sonication. So the question remains: Are there additional concentrations of fatty acids after base treatments which show up later in other fractions?
Author Response
Reviewer 1 - Round 2
It is quite clear that a base treatment before lipid extraction and analysis makes less sense. What is meant is: we know diglycerides bound to cell walls via sugars which are insoluble. This cannot be extracted by simple solvent extraction and/or sonication. So the question remains: Are there additional concentrations of fatty acids after base treatments which show up later in other fractions?
Thank you very much for taking the time to review this manuscript. Please find the detailed responses below and the corresponding revisions/corrections highlighted/in track changes in the re-submitted files.
Response: In the present study, our objective was to investigate the soluble phase. For this reason, the pellet fractions were not retained or analyzed in our experimental design.
However, we fully acknowledge that the insoluble residues, including the pellets, may contain lipid fractions of interest particularly those associated with cell wall or membrane structures. This is indeed a valuable consideration, and we see it as a promising approach for future investigations aimed at achieving a more comprehensive understanding of the lipid profile. We changed the title of the methodology section to clarify that the analysis was done without alkaline hydrolysis: ‘General characterization prior to alkaline hydrolysis optimization’.

Round 3
Reviewer 1 Report
Comments and Suggestions for Authors
x